# Large gaps in monitoring urban air pollution in low- and middle- income countries associated with economic conditions and political institutions

Maja Schoch☯, Camille Fournier De Lauriere☯, Thomas Bernauer🆔*

Institute of Science, Technology and Policy (ISTP), ETH Zurich, Haldeneggsteig, Zurich, Switzerland

☯ These authors contributed equally to this work.
* thbe0520@ethz.ch

## Abstract

Ambient air pollution has highly adverse effects on public health and the environment, particularly in urban areas of low- and middle- income countries (LMIC). Systematic air quality monitoring (AQM) is considered a precondition for effective policies to mitigate this problem, and making AQM data publicly available also signals commitment to take action. Thus far, little is known about the global capacity for public AQM, and how it varies across geographic location, pollution exposure, and socio-economic and political conditions. We thus constructed a novel, geocoded dataset on AQM activity in more than ten thousand urban areas of low- to middle-income countries. In almost 90% of these urban areas, we are unable to identify any monitoring activity, and the form and extent of AQM in the remaining 10% varies greatly. When modelling the occurrence and abundance of observable AQM at the city level, income levels and characteristics of political institutions (democracy) turn out to be key drivers of variation in AQM, with urban areas in more democratic countries likely to exhibit more AQM when air pollution levels are high. The evidence provided here could motivate public authorities, international institutions, and civil society stakeholders to invest far more than hitherto the case into AQM, particularly in under-monitored, less affluent, and less democratic settings.

## Introduction

Ambient air pollution has deleterious effects on human health, the environment, and the economy worldwide (e.g. [1–6]). Urban areas in low- to upper middle-income countries, which we refer to as LMIC, face particularly serious problems in this realm due to rapid population growth and expanding industrial activity [7–9]. For instance, according to the Health Effects Institute [10], air pollution is estimated to be the second leading risk factor for death in Africa, following malnutrition, with more than one million estimated deaths in 2019 and around 14 percent of child deaths under age

**Data availability statement:** All code produced to create figures in the manuscript is available here: https://github.com/camillefournierdl/AQMdeterminants. All data files are available from public databases, with the exception of commercial data from PurpleAir. We thank all other providers of publicly available data used in this analysis, notably WAQI and OpenAQ for making their data available to all.

**Funding:** Swiss National Science Foundation, Project Number 10521G_219833.

**Competing interests:** No author has competing interests.

five linked to air pollution. These problems are amplified by the fact that urban air pollution coincides and interacts with climate change and other ecological stressors, which implies that substantial synergies and co-benefits could be achieved from pursuing both clean air and climate mitigation measures [11–13]. These challenges call for more air quality monitoring (AQM) to identify hotspots and design, implement, and evaluate clean air policy interventions.

It is widely acknowledged that systematic air quality monitoring (AQM) is indispensable for effective clean air policy [14]. Yet, the apparent shortage of data hampers not only problem-solving efforts upfront. It also precludes continuous evaluation and improvement of existing intervention strategies. Conversely, AQM that generates reliable, publicly available information could enhance public awareness and support stronger action against air pollution [15,16]. As suggested by one recent study [16], the presence of AQM at US embassies in more than 40 cities may have contributed to lower air pollution levels in the respective cities. And as succinctly noted by Harrington: "*Measurement is the first step that leads to control and eventually to improvement. If you can't measure something, you can't understand it. If you can't understand it, you can't control it. If you can't control it, you can't improve it.*" [17]. Several studies observe, moreover, that policymakers gain or lose public support because of improving or deteriorating air quality, thus underscoring that AQM is closely tied to processes of political accountability (e.g. [18,19]).

It is worth noting in this context that remote sensing data on air quality are becoming more widely available, e.g. for fine particulate matter ($PM_{2.5}$). However, their spatial and temporal resolution is usually too coarse to support the design and implementation of well-targeted local clean air policies [20–23]. Recent discussions suggest, however, that AQM networks should incorporate both remote and on-the-ground monitoring (AQM), to make the most of available data sources [24,25]. Existing literature points to important monitoring gaps in LMIC and suggests reasons for such gaps. But it does not yet offer a systematic characterization of spatial patterns in AQM (e.g. [2,7,26,27]). As a consequence, systematic analysis of political, socio-economic, and other drivers of variation in AQM is also missing.

Here, we characterize spatial patterns of AQM throughout LMIC and identify potential drivers of variation in AQM, focusing on thousands of urban areas across low- to upper middle-income countries, rather than countries as such. The reason is that aggregating AQM activity to the country level would blur differences between and within countries. Our analysis focuses on three key drivers of environmental policy preferences, which are also presumed to influence behaviour in the area of AQM. These are economic resources, political institutions, as well as environmental problem pressure, and responsiveness to it.

Economic resources are widely regarded as a key enabler of more stringent environmental policies and, as a corollary, of more environmental monitoring [16,28–30]. According to the hierarchy of needs argument [31], higher average income levels eventually induce more public demand for cleaner air, and by implication also more demand for measuring air quality. Moreover, higher income levels go hand in hand with more resources that are potentially available for AQM. Both mechanisms are deeply intertwined and thus hard to separate analytically because higher income

levels are likely to induce both more demand for AQM and capacity to meet this demand. One noteworthy recent development is that technological innovations are offering new opportunities for low-cost monitoring, which is especially relevant in lower-income contexts, though much more expensive reference-grade monitors are still considered the 'gold standard' in AQM both for reliability and regulatory/legal reasons [32]. Consequently, we distinguish between the two types of AQM, expecting monitoring activity to increase with income levels.

Previous studies highlight the impact of democratic institutions on environmental policy choices and their outcomes. They provide both theoretical arguments and empirical evidence in favor of the expectation that democratic political institutions make societies more likely to implement more ambitious environmental policies and achieve higher levels of environmental system quality [33–36]. On the demand side, democracy makes it easier for scientists, citizens, civil society, and other stakeholders to identify environmental problems and aggregate and organize various demands into ways and means that put pressure on policymakers to act. On the supply side, in democracies, policymakers have stronger incentives to meet the demands of a broad range of citizens, relative to autocracies that tend to be governed by a small elite [34]. The key reason is that democratic policymakers need to be (re-)elected [37–40]. We thus expect that demand and supply mechanisms are, jointly, likely to result in a positive effect of democratic institutions on AQM.

As a corollary to the democracy argument, we also expect that urban areas are more likely to respond to higher pollution levels with more AQM in settings characterized by more democratic institutions. The reason is that, as air pollution increases, the issue is more likely to become politically salient in a democratic setting, and there is thus likely to be more pressure on policymakers to offer solutions to the problem. A first step towards such solutions is usually more AQM.

## Data and methods

### Unit of analysis

To characterize the spatial distribution of monitoring activity, we construct a new dataset by geo-matching the location of monitoring stations from which air pollution data is reported with the coordinates of so-called Urban Centres (UC). These are urban centers housing a population of at least 50'000 and having a population density of 1'500 or more per square kilometer [41,42]. We use UC and the term city synonymously throughout the paper.

Even though air pollution is widely regarded as having disproportionately negative effects on people in low-income countries [43], current studies on clean air policies are strongly biased towards high-income countries, underscoring the need for a comprehensive analysis of lower-income nations [14,44] where incomes are, on average, around four times lower than in high-income countries [45]. We therefore exclude UCs in high-income countries.

The complete dataset includes 11'106 UCs, most of them located in India or China (see Figs 1 and 2). To obtain a balanced dataset and avoid over-representing these two countries, we randomly subsample UCs from these two countries, resulting in 500 cities each for China and India, matching Ethiopia, the third-largest country in terms of UCs. This process yields a final dataset of 7'106 cities, of which 1'365 have at least one monitoring station. As a robustness check, we use different samples from the full dataset as well as the full dataset including all cities in China and India.

### Data sources

**Monitoring stations.** With UCs as the unit of analysis, we merge data on reported terrestrial AQM activity from three different sources with global coverage, monitoring any activity up to mid-April 2024. The databases differ in monitoring instruments, including (low-cost) air sensors and reference monitors, and coverage focus [47–49]. The inclusion of all three sources ascertains that we cover different parts of the world as well as different monitoring systems. While the Purple Air source focuses on low-cost sensor monitoring stations, the WAQI and OpenAQ include mostly reference-grade monitoring stations, (see Figs. S6 and S7).

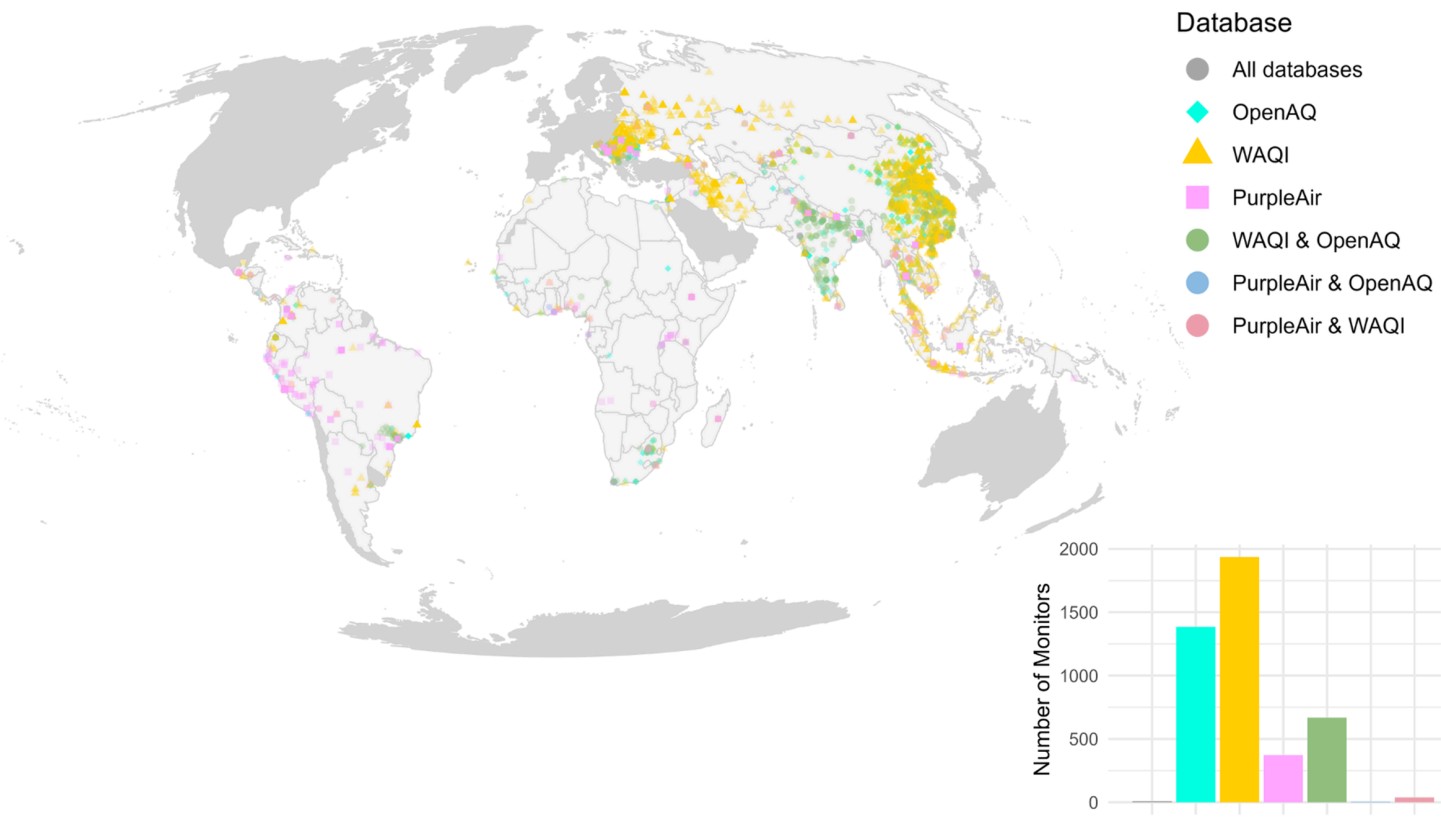

**Fig 1**. **Highly uneven geographic distribution of AQM.** Points on the map represent locations of AQM provided by the three data platforms we consider in our analysis (OpenAQ, WAQI, and PurpleAir). The histogram shows the number of monitors and overlaps for coverage by the three data platforms. Each data source is displayed using a different shape, and any AQM location that overlaps in two or all data sources is displayed as a circle marker. Countries excluded from the analysis (high-income countries) are coloured in grey. The country map was downloaded from the Natural Earth R package [46].

Not all AQM activities show up in OpenAQ, WAQI, or PurpleAir. For example, companies, local AQM campaigns, or specific countries might opt not to make their data publicly available. Furthermore, OpenAQ and WAQI are continuously adding new public reports of air pollution data, which need to be machine-accessible, adding technical barriers to what kinds of data are being ingested in these databases. Also, OpenAQ ingests data in raw units (e.g. µg/m$^3$ for PM$_{2.5}$), whereas WAQI compiles different pollutants into a comprehensive Air Quality Index, and can ingest such indices, which explains why these datasets offer different spatial coverage. Differences in what gets added into OpenAQ and WAQI, as well as what is available in the PurpleAir database led us to consider any AQM reporting in our analysis: various raw pollutants and Air Quality Index-related data. This allows us to cover any public AQM, which is what "ordinary" citizens would access to inform themselves about the air quality where they live. 68 monitors were placed at US embassies. We exclude them from the main analysis because the installation of such monitors is exogenous to the drivers we wish to explore in our analysis. Unfortunately, these datasets do not always add enough information about the monitors, to assess whether they are governmentally or privately, or NGO operated. We expect most reference-grade monitors to be installed with the approval of the local or national government, while low-cost sensors could be set up by any actor, including governmental entities.

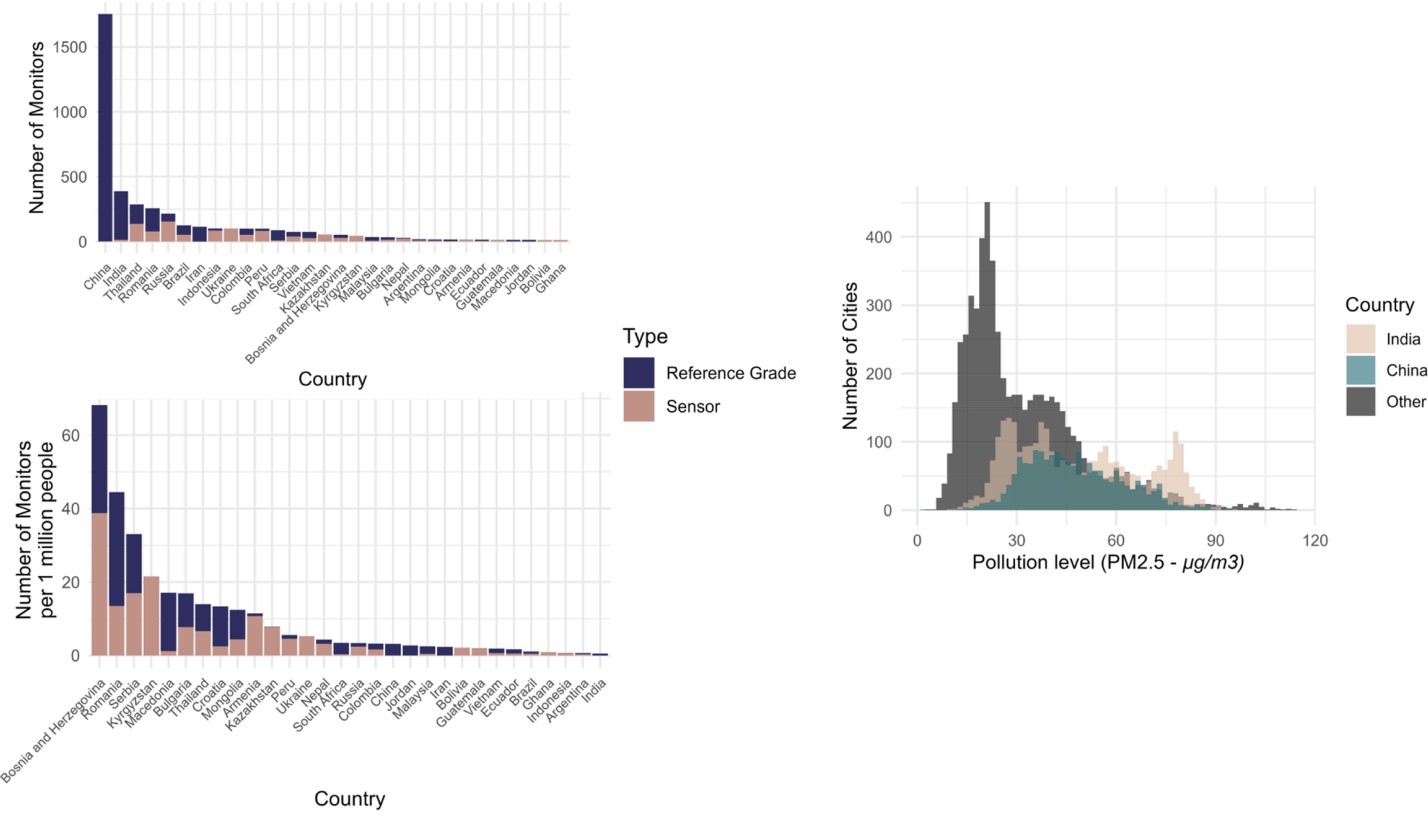

**Fig 2**. Number of reference-grade and sensor-based (i.e. low cost) monitors per country (upper left) and per country normalized by population size, and number of cities with AQM in China, India, and other countries by pollution level.

Finally, data providers like WAQI and OpenAQ use data validation algorithms to identify outliers and abnormal measurements. As an additional precaution, we exclude monitors that were reporting for less than 30 days, and those that reported for less than 2 weeks on the WAQI API in April 2024.

**Economic resources.** Economic resources are measured using the Gross Domestic Product (GDP) per capita (p.c.) of each UC for the year 2015, as provided in the OECD dataset. The database enumerates GDP estimates computed using global figures for the annual total GDP based on purchasing power parity (PPP) within the Urban Centre 2015, denominated in US dollars (base year 2007). As these figures are accessible at a 30 arc-second resolution (about 1 km at the equator), it enables us to use information on GDP for each city and not only on a standardized country level [50]. To capture the prosperity of an UC, not influenced by its size, the variable is divided by the population size in the same year (2015), which results in the GDP p.c. of an UC. To achieve a normal distribution of the variable, we use the logarithm of the GDP p.c. in our analysis.

**Political institutions.** The measure for democracy employed in our analysis is the Electoral democracy index by V-Dem. It consists of five sub-components that together capture Dahl's seven institutions of polyarchy: freedom of association, suffrage, clean elections, elected executive, freedom of expression, and alternative sources of information [51]. The resulting index ranges from 0 (low performance) to 1 (best performance). We average V-Dem values between 2000 and 2015 and consider a country as democratic when the V-Dem Electoral democracy index is > 0.5, and as non-democratic if its rating is equal to or below that value.

 

**Air pollution.** To measure air pollution, we use remote-sensed estimates of $PM_{2.5}$ concentrations, expressed in $\mu g/m^3$ as the total concentration of $PM_{2.5}$, averaged for every UC during the 2000-2016 period. This provides us with an exogenous measure of air pollution and a consistent scale around the globe. These remote-sensed estimates of $PM_{2.5}$ are built from models that rely on ground measurements, and thus carry more uncertainty in regions that do not have terrestrial AQM. The uncertainty in remotely sensed pollution should not lead to a systemic bias for cities that do not report on-the-ground measurements [52]. However, because we average values over 17 years, we minimize potential biases from monthly measurements. This is, to our knowledge, the most adequate (and mostly exogenous to on the ground AQM) data source of air pollution available for our study. It covers all UCs in our dataset except for two Russian cities.

**Conflict.** To measure conflict, we use data from the UCDP/PRIO Armed Conflict Dataset, version 23.1 [53,54]. This dataset offers a range of information about conflict, including aspects such as intensity, conflict type, and the start and end date of the conflict. Given that a conflict can significantly shape the development of a whole country and considering the complexities of aligning the conflict location with the UC dataset, we opt to incorporate the conflict variable at the national level and use it as a dummy variable that identifies if a conflict has resulted in over 1'000 battle-related fatalities since its inception. As a country is also highly affected by the aftermath of war, the main variable considers any onset of a war from 2000 until 2022.

**Corruption.** To measure corruption within a country, we rely on the Corruption Perceptions Index (CPI), created by Transparency International [55] and use the average levels between 2012 and 2022. The CPI ranks countries based on perceived public sector corruption, aggregating data from different sources that reflect expert and business evaluations of public sector corruption. The CPI is standardized on a scale from 0 to 100, ensuring year-to-year comparability. While within the index itself, a lower score signifies higher corruption and a higher score indicates lower corruption, we adjust the variable so that a higher value denotes increased corruption, and a lower value signifies reduced corruption. This makes the interpretation easier and simplifies comparisons across variables.

**Other control variables.** For control variables, we included the 2015 population size of each UC as an indicator of urbanization [56]. Due to the significant skewness of this variable, we apply a logarithmic transformation before inclusion in the regression models. Additionally, we included a dummy variable that denotes whether a UC is the capital of a country [56]. This variable is important as it is likely that many (monitoring) policies are first rolled out in a country's capital before being adopted in other urban areas. Additionally, we include dummy variables for China and India to control for fixed effects within the many UCs in these countries.

## Data analysis

To explore the determinants underpinning monitoring behavior and address the outlined hypotheses, the initial approach uses bivariate analyses. This involves comparing mean values across distinct groups, and testing significance with permutation tests because firstly, visual inspection of the data revealed non-normality in the distribution of the groups, and secondly, unequal sample sizes can bias Welch tests towards lower p-values [57]. We used the perm library [58] with a Monte Carlo Approximation using 1'000 permutations. This approach is robust to non-normality and differences in sample sizes between two focal groups. To account for all theoretically defined factors discussed earlier, including interactions between these factors, we employ two multivariate analyses. We model the presence (or absence) of at least one monitoring station within an UC using binomial logistic regression (Eq (1)). We also model the number of monitors in UCs, using a Poisson regression model [59], because of the zero-inflated nature of our dataset, with most UCs not reporting AQM (Eq (2)).

$$logit[Pr(\text{Presence of AQM}_{it} = 1)] = \beta_0 + \beta_1 \log(\text{GDPpc}_{it}) + \beta_2 \text{Democracy}_{it}$$
$$+ \beta_3 \text{PM}_{2.5,it} + \beta_4 \text{PM}_{2.5,it} \times \text{Democracy}_{it}$$
$$+ \beta_5 \log(\text{Population}_{it}) + \beta_6 \text{Conflict}_{it}$$
$$+ \beta_7 \text{Corruption}_{it} + \beta_8 \text{Capital}_{it} + \beta_9 \text{India}_i + \beta_{10} \text{China}_i \tag{1}$$

$$\log(\mathbb{E}[\text{Number of Monitors}_{it}]) = \beta_0 + \beta_1 \log(\text{GDPpc}_{it}) + \beta_2 \text{Democracy}_{it}$$
$$+ \beta_3 \text{PM}_{2.5,it} + \beta_4 \text{PM}_{2.5,it} \times \text{Democracy}_{it}$$
$$+ \beta_5 \log(\text{Population}_{it}) + \beta_6 \text{Conflict}_{it}$$
$$+ \beta_7 \text{Corruption}_{it} + \beta_8 \text{Capital}_{it} + \beta_9 \text{India}_i + \beta_{10} \text{China}_i \tag{2}$$

To assess the robustness of the results, we use different combinations of decisions and assumptions throughout the analysis and test them using multiverse analysis [60]. The multiverse package is useful for adding transparency regarding methodological decision-making and its impact in the analysis. The package offers a syntax that allows scientists to test consistency when using different combinations of methodological strategies, in an easy-to-test and report syntax, instead of reporting one of them. While for the main analysis, we use a sub-sample for China and India, 500 UCs each, we also run the regression analysis using the complete dataset that includes all UCs in China and India. In another regression, we exclude PurpleAir AQM stations, which focus extensively on low-cost air sensors, thus the remaining dataset mainly contains reference-grade monitors. For another robustness check, we run the analysis with a second pollution dataset, the UC-level pollution dataset from "Urban Centre spatial domain based on Global Burden of Disease (GBD) 2017", derived from older remote-sensing estimates [52]. Finally, we also run our analysis by including US embassy monitors. As S1, S3 and S4 Figs show, none of these adjustments strongly influence our main finding.

We also test the sensitivity of findings to the random sub-sampling that we use to reduce the overrepresentation of China and India. To do that, we repeat our analysis 1'000 times, each time selecting a different sample of the 500 UCs in China and India. We plot the coefficients of the interaction between pollution levels and non-democracy in S5 Fig.

As we find that democracy influences AQM at the city level and democracy is a country-level variable, we model the number of monitors also at the country level using a Poisson regression. To do that, we use the country level variables such as corruption, conflict, and democracy, and summed up the total number of monitors in UCs for each country, as well as the total population living in UCs, the GDP p.c. at the country level, for the population in UCs, and the average pollution in UCs, for every country included in our analysis. The results show that our findings are robust at the country level, with more democratic countries having more publicly reporting monitoring stations with higher pollution, while non-democracies have more AQM if when pollution levels are lower.

## Results

AQM activity in the cities of interest here is unevenly distributed around the world, with high monitoring density in China and India, and very low monitoring density in Africa and South America (Fig 1). The three data platforms from which we extract locations of AQM provide varying coverage across different regions of the world, making it useful to include all three in our analysis (see Data and methods). There is little overlap between data from the air sensor provider PurpleAir and the two other data sources that focus more on Eastern Europe, China, and India.

China and India account for more than 2'100 and 380 monitors respectively, and over 1'800 and 3'100 cities respectively, with some cities among the most polluted globally (Fig 2). This high concentration of AQM in China and India, and

the high share of reference-grade (as opposed to low-cost sensor) monitoring in these countries requires special attention when exploring drivers of variation in AQM (see Data and methods). However, it is also worth noting that China and India only rank 18th and 30th when normalizing the amount of AQM by urban population (Fig 2). Fig 2 also shows that a large share of AQM in China and India takes place in cities experiencing medium to high levels of air pollution (remotely sensed). A much larger share of AQM in cities of other countries focuses on areas with relatively low to medium pollution levels (again remotely sensed). Again, this implies that we need to pay special attention to China and India when exploring the drivers of AQM, most notably how political systems and problem pressure act in combination.

We now move to identifying the conditions under which we are likely to observe AQM when comparing cities across LMIC. Fig 3 offers some first, bivariate insights. We are more likely to observe AQM in cities with higher income levels (p < 0.01, permutation test, log difference = 1.40), while AQM activity does not seem to be associated with the level of democracy (p > 0.18, permutation test, difference = 0.01). Cities with lower pollution levels are more likely to have air quality monitoring (p < 0.01, permutation test, difference = 3.93). Comparing political contexts, cities in non-democratic settings tend to be more polluted than those in democratic settings (p < 0.01, permutation test, difference = 8.25).

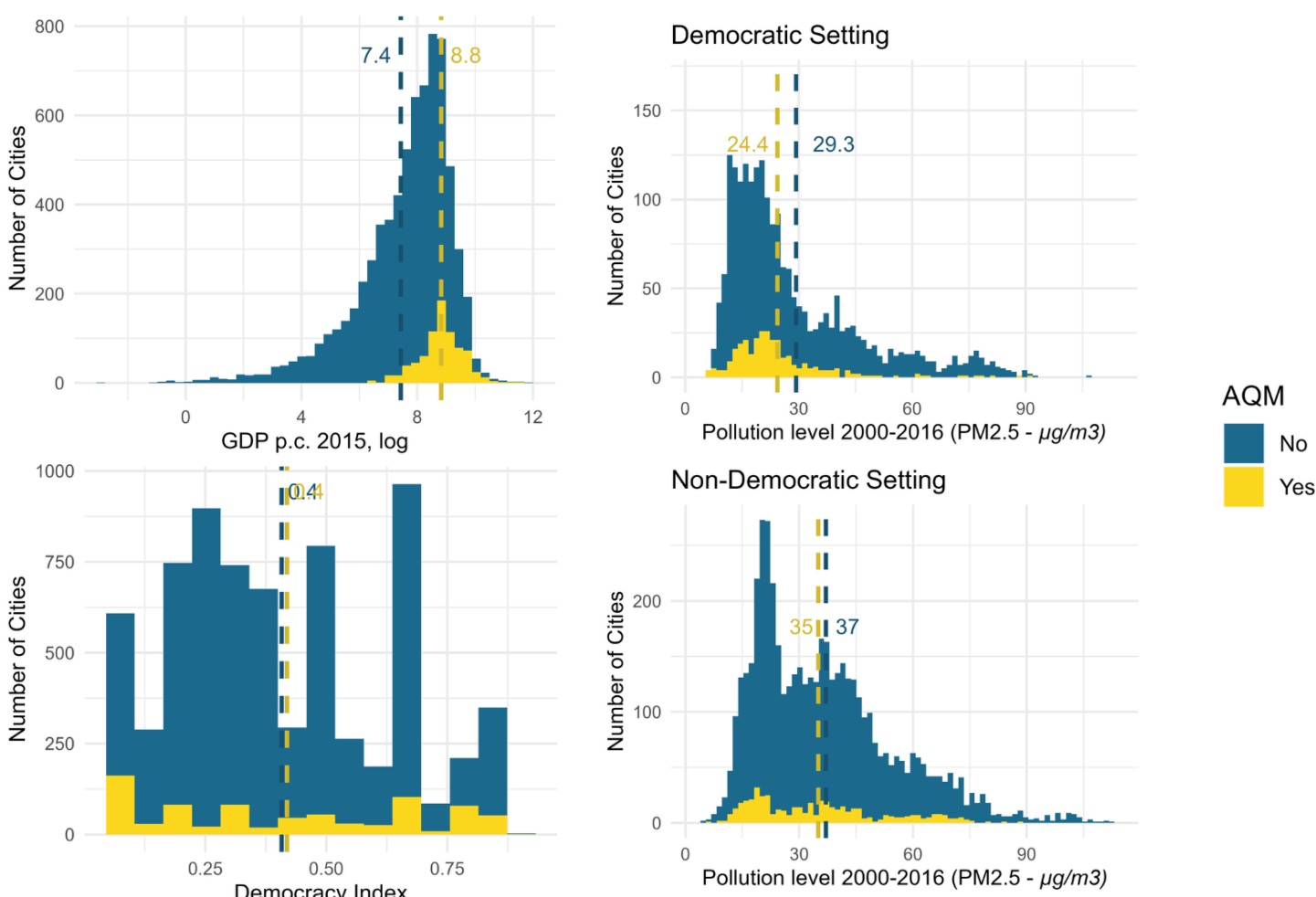

**Fig 3. Potential drivers of variation in AQM activity.** Dashed lines indicate the mean values for the explanatory variable displayed in each graph, for cities with AQM (yellow) and those without (blue).

However, within both regime types, cities that have air quality monitoring (AQM) are significantly less polluted on average, with the difference being larger in democratic cities (p < 0.01 for both, permutation tests, difference of 2.01 and 4.97, respectively). This observation aligns with the assertion that cities in democratic settings exhibit more AQM when pollution levels are higher.

To better understand structural facilitators and obstacles to AQM, we control for confounders based on a regression analysis. Fig 4 shows predicted effect sizes for the main variables of interest (left side), and for a set of other factors (right side) frequently referred to in the literature on AQM.

The results indicate that AQM is – all else equal – more likely to occur in cities that are (within the income range of the countries we consider) wealthier, located in more democratic countries, and have higher pollution levels, particularly in democracies. This means that more polluted cities in democracies are more likely to have AQM than similarly polluted cities in non-democracies. Cities in countries where there is a war also exhibit a higher likelihood of AQM. Larger populations and capital city status increase the probability of AQM as well. Conversely, high corruption levels reduce the likelihood of AQM. The results differentiated for reference-grade and (low-cost) sensor-based AQM are shown in S1 Fig.

Adding further evidence for the argument about democracy effects, Fig 5 shows how the predicted probability of AQM being present changes with increasing pollution levels for cities in democratic and non-democratic countries (pollution levels are again captured with remote sensing data to avoid endogeneity bias). S2 Fig shows the same interaction effect within the regression model for different economic contexts. It shows that this interaction effect materializes mainly in economic contexts other than very low- and very high-income settings. Irrespective of democracy, very poor cities have a very low predicted probability of AQM being present, and very rich cities in our sample have a very high probability, even if they are moderately polluted.

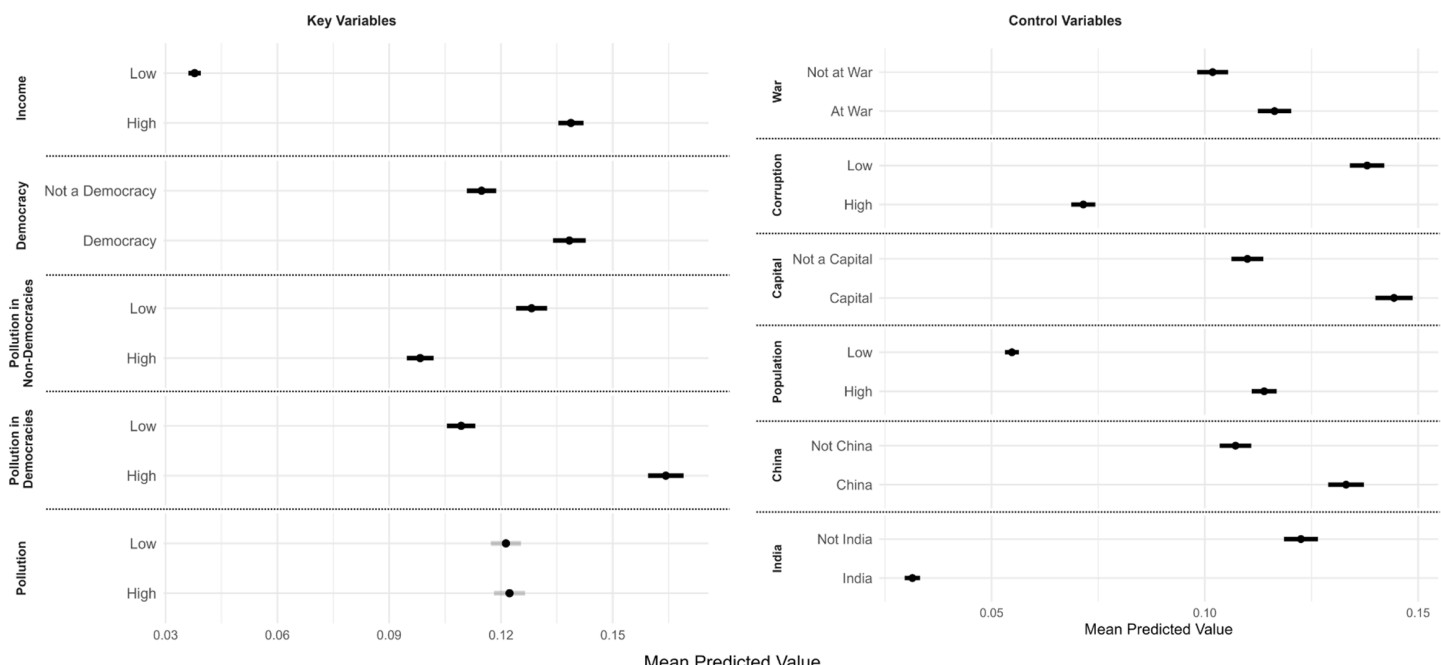

**Fig 4. Effects of explanatory variables.** Values closer to 1 mean a strong predicted probability for the city to report AQM data. The marginal effects are calculated by fixing a variable at a specific value and running predictions in the model, keeping all other variables at their observed average values. Marginal effects for continuous variables are calculated at the first (in the figure referred to as 'low') and third (referred to as 'high') quartiles of the observed distribution. Categorical variables' marginal effects are calculated for both levels of each variable (e.g. 'At war' vs 'Not at war'). The model building the ground for this figure is a binary logit regression that includes the variables listed in the figure, see Eq (1).

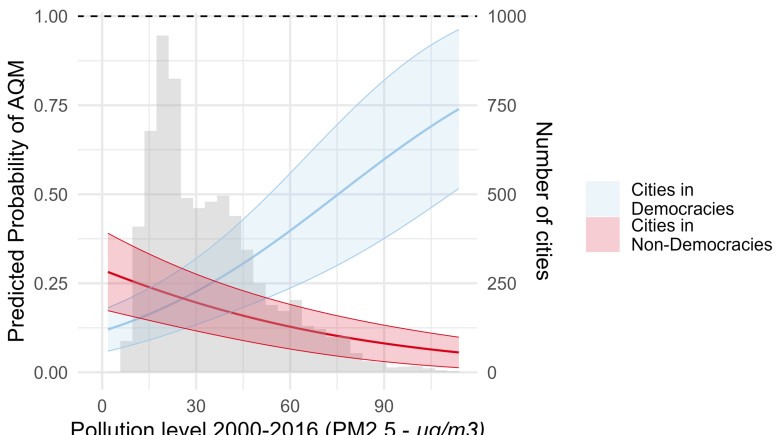

**Fig 5. Predicted probability of AQM relative to pollution levels (PM₂.₅) from 2000-2016 in cities located within a democratic or non-democratic setting.** The model pictured here is the same binary logit regression model as in Fig 4. It includes all variables described in the methods part. The variables not shown in the graph are set to the mean of the observed values. The grey histogram shows the distribution of the pollution levels across the cities included in the regression.

We examined the robustness of our main findings in several ways. These include using a different dataset for remotely sensed air pollution ("Urban Centre spatial domain based on Global Burden of Disease (GBD) 2017 data"), different random sub-samples of cities in China and India, data for all cities including cities in China and India, excluding AQM activity captured with data from air sensors of PurpleAir, and including or excluding AQM by US embassies (Fig 6 and S3 Fig).

The results indicate that the presence of AQM is associated with higher pollution levels in cities within democratic settings. Although the effect is weaker for reference-grade monitors, it remains statistically significant when applying our baseline model (Fig 6). The effects remain significant when using a Poisson regression to predict the number of monitors (rather than the presence or absence of AQM), accounting for the zero-inflated distribution of AQM, with robust results across different regression models and datasets (S3 Fig). Because democracy is ultimately a country-level variable, we also conduct a Poisson regression analysis to predict the number of monitors in each country based on the average income level, population, and air pollution levels of all cities in a country. We find similar results with weaker statistical significance (S4 Fig and S1 Table). Finally, we examined whether the presence of US embassy-based AQM may crowd in or crowd out other AQM in the respective city. We find that when controlling for other factors, such as income, population, and democracy, there are fewer reference-grade monitors, and more air sensors, in capital cities that host a US embassy monitor, but find no effect when not differentiating monitor type (S2 Table).

Calculating Moran's I to measure residual autocorrelation after accounting for city and country-level determinants, we find that cities within a 100 km radius from each other are more likely to exhibit the same monitoring outcome (Moran's I = 0.26, p < 0.01), see Supplementary Analysis in Appendix.

## Discussion

Air pollution constitutes a major public health challenge worldwide, particularly so in urban areas of LMIC. Air quality monitoring is important to making progress towards cleaner air. Remote sensing data on air quality is increasingly available, but ground-level estimates of air pollution levels still need to be calibrated with data from on-site monitoring, and regulatory policy in this domain commonly must rely on reference-grade monitors for legal and political legitimacy reasons. Moreover, the spatial and temporal resolution of existing remote sensing data on urban air pollution is still too coarse for

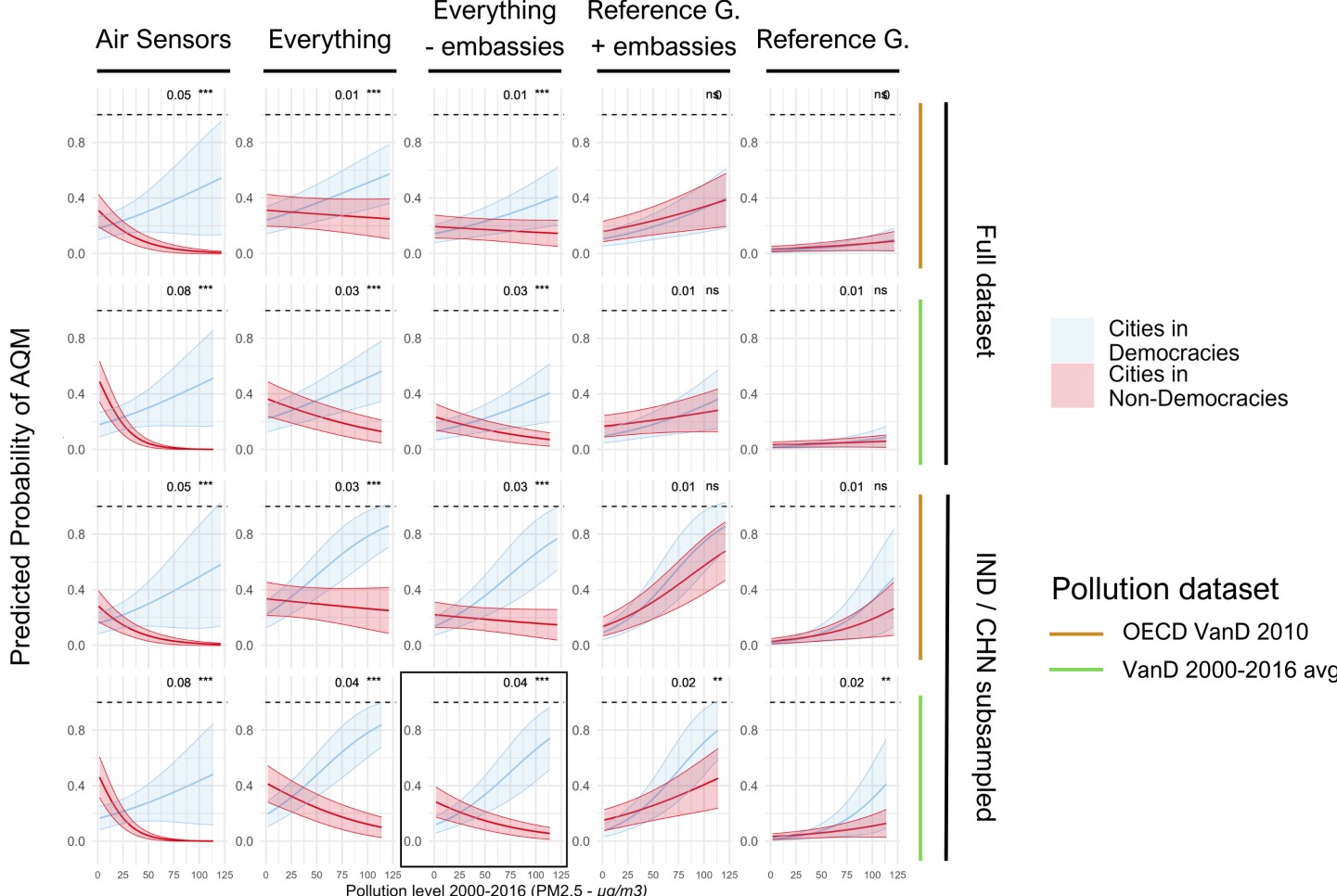

**Fig 6. Robustness of main results.** This figure shows that the main pattern of how AQM activity behaves with increasing pollution levels in democratic and non-democratic settings remains similar when using different data sources for remotely sensed pollution, including all cities or downsized samples for China and India, including only reference-grade or low-cost air sensor AQM or both, and including or excluding AQM by embassies. Numbers on the top right of each graph display the effect size of the interaction term between democracy and pollution, along with its significance level. The graph in the black frame corresponds to the main analysis that is displayed in Fig 5.

localized action in terms of identifying pollution hotspots within cities, building public awareness, enforcing regulations, tracking progress, and ultimately reducing pollution levels. This means that cities cannot rely on remote sensing data alone but must invest in localized AQM too, though exploiting synergies between local AQM and remote sensing data remains crucial. It is noteworthy in this context that remote sensing is gaining in importance, with governments blending satellite data, legacy reference monitors, and low-cost sensors. Yet reference-grade monitoring remains essential for calibration and enforcement [25].

The present paper is, to our knowledge, the first to describe variation in AQM across all cities in LMIC and explore potential drivers of such variation. Besides observing a glaring gap in AQM across a vast part of cities of LMIC, our analysis highlights economic and political conditions as key drivers of variation in AQM. While the positive effect of income levels on AQM probably lines up with common intuition, the fact that the association of increasing pollution levels with AQM is contingent on democracy is in our view quite intriguing. All else equal, public authorities in more democratic

settings appear to be more responsive to increasing pollution levels, probably both by engaging and allowing others to engage in more reference-type and low-cost sensor-based AQM.

Differences between reference-grade and low-cost sensors suggest that in non-democracies, governments set up one reference-grade station (the reason for that could be studied in further research) and commonly leave it at that, whereas low-cost sensors proliferate particularly in highly polluted areas of the more democratic LMIC.

The observation that non-democracies are more likely to monitor in lower pollution areas suggests that non-democracies' monitoring locations may be biased by political considerations. While non-democracies may establish monitors in cleaner places in response to international pressures, democracies may prioritize highly polluted cities and establish more extensive monitoring networks, due to electoral incentives to provide public environmental goods. Although environmental agencies in democracies generally operate with greater autonomy, instances of strategic misreporting have been documented in both the US and China [61,62].

The findings presented here could encourage public authorities, international institutions, and civil society stakeholders to invest far more than hitherto the case into AQM, particularly in under-monitored, poorer, and less democratic settings, drawing on lessons learned from Bagkis et al. [24] and Anggraini et al. [63]. Country-external governmental or non-governmental actors could catalyze more AQM via technical assistance (e.g. calibration, open-data systems), co-financing of reference backbones and low-cost densification of monitoring networks. They could also create more pressure on policy-makers to make monitoring results public and allow for independent audits. One interesting hypothesis to study in further research could be whether cities in more democratic contexts are more likely to receive such assistance and make more efficient and effective use of it.

Our research also points to other interesting avenues for further research. We analyze city and country level determinants, but a superficial analysis of spatial autocorrelation suggests that cities within about 100 km of each other tend to share similar likelihoods of having monitoring stations, highlighting the role of regional governance. Further, focusing in greater depth on the cities with (currently) observable AQM could provide interesting insights into variation in the spatial and temporal density as well as pollutant coverage of monitoring, variation in how public authorities and other actors are making pollution data public, and which efforts are made (or not made) to improve monitoring networks. Such work could also examine whether there are socio-economic biases in AQM, e.g. more monitoring in more affluent areas of a city, and vice versa. This would require finer-grained data than what was used in this paper to capture within-city variations, and could be coupled with qualitative interviews to describe the process of AQM placement. It could also study the conditions under which improvements could be achieved with low-cost sensors that are operated by both governments and independent actors (private individuals, NGOs, companies engaging in transparency). Experimental study designs, such as RCTs randomizing the placement of low-cost monitors could help understand the impact that more local air quality information may have on people's awareness, concern, behavioral changes to reduce pollution exposure, and policy-preferences and demands.

In identifying AQM activity, further research could also investigate potential biases when picking up AQM from reported pollution data. If, as one might suspect, non-democratic regimes measure air pollution, but withhold the data from the public to avoid criticism, this could explain why cities in democratic settings seem to be more responsive to high levels of air pollution. The fact that our data identifies AQM activity in non-democratic settings (above all China, where we observe a vast amount of AQM) implies that such bias is probably limited. Nevertheless, more work is needed to understand underlying mechanisms leading to the limited extent of AQM in some of the most polluted non-democracies.

Further, our results indicate that India and China diverge notably from other LMIC cities in their AQM coverage, reflecting different histories regarding air pollution management. China's nationwide AQM campaign led to a dense monitoring network [64], compared to other less-democratic countries experiencing high pollution. In contrast, at our cut-off in April 2024, India's monitoring infrastructure remained highly concentrated in Delhi [65], making other cities in India less likely to have AQM compared to other polluted cities in democratic settings. Previous research in these countries is primarily descriptive, historical, or seek to identify whether more monitoring coincides or is followed by reduced air pollution. It do

not systematically explain variation in AQM across cities, for instance as a function of income levels and exogenously measured pollution. They could serve as a foundation for such work, though variation in AQM across political systems cannot be studied within countries. More research is needed to better understand why these countries behave as outliers compared to the other LMIC when looking at the role of political systems. Future work could also examine how platforms like OpenAQ, WAQI, and PurpleAir are improving the transparency and robustness of their data quality practices, including outlier detection and handling.

While action to reduce air pollution exposure is urgent, and policies may not always necessitate a large network of AQM [66], more information can help create pollution alert systems [67,68], and help raise public awareness and support for policies aimed at tackling air pollution.

As such, we believe that AQM and environmental monitoring in other domains for that matter are key elements in societal efforts aimed at improving environmental conditions and deserve more scientific attention than is currently the case. We thus hope that the present paper encourages others to explore further the inferences we can draw from reported environmental data for the preferences and behavior of public authorities and other stakeholders.

## Supporting information

**S1 Appendix. This Appendix includes additional figures and tables referenced in the manuscript, including additional analysis such as a country-level Poisson regression, the effect of US-embassy monitors on other AQM, & sensitivity tests such as robustness to subsampling India and China, to different methodological decisions on results from Eq (2), and to using a different data source for income.**
(PDF)

## Author contributions

**Conceptualization:** Maja Schoch, Thomas Bernauer.

**Data curation:** Maja Schoch, Camille Fournier De Lauriere, Thomas Bernauer.

**Formal analysis:** Maja Schoch, Camille Fournier De Lauriere.

**Funding acquisition:** Maja Schoch, Thomas Bernauer.

**Investigation:** Thomas Bernauer.

**Methodology:** Maja Schoch, Camille Fournier De Lauriere, Thomas Bernauer.

**Project administration:** Thomas Bernauer.

**Supervision:** Thomas Bernauer.

**Visualization:** Maja Schoch, Camille Fournier De Lauriere.

**Writing – original draft:** Maja Schoch, Thomas Bernauer.

**Writing – review & editing:** Maja Schoch, Camille Fournier De Lauriere, Thomas Bernauer.

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
