## [Decision Letter · Decision Letter 0]

28 Aug 2025

PONE-D-25-11715Monitoring Urban Air Pollution in the Global South: Large Gaps Associated with Economic Conditions and Political InstitutionsPLOS ONE

Dear Dr. Bernauer,

Thank you for submitting your manuscript to PLOS ONE. After careful consideration, we feel that it has merit but does not fully meet PLOS ONE’s publication criteria as it currently stands. Therefore, we invite you to submit a revised version of the manuscript that addresses the points raised during the review process. The reviewers are positive about the manuscript but find some elements in different parts of the article that can be improved. Please carefully consider the different points made by reviewers (especially the ones on data and methods) and revise the manuscript accordingly.

We look forward to receiving your revised manuscript.

Kind regards,

Floris Vermeulen

Academic Editor

PLOS ONE

“Swiss National Science Foundation, Project Number 10521G_219833.”

6. We notice that your supplementary tables and Figures are included in the manuscript file. Please remove them and upload them with the file type 'Supporting Information'. Please ensure that each Supporting Information file has a legend listed in the manuscript after the references list.

Reviewers' comments:

Reviewer's Responses to Questions

**Comments to the Author**

1. Is the manuscript technically sound, and do the data support the conclusions?

Reviewer #1: Yes

Reviewer #2: Yes

Reviewer #3: Yes

Reviewer #4: Yes

Reviewer #5: Yes

2. Has the statistical analysis been performed appropriately and rigorously?

Reviewer #1: Yes

Reviewer #2: Yes

Reviewer #3: Yes

Reviewer #4: Yes

Reviewer #5: Yes

3. Have the authors made all data underlying the findings in their manuscript fully available?

Reviewer #1: Yes

Reviewer #2: Yes

Reviewer #3: Yes

Reviewer #4: Yes

Reviewer #5: No

4. Is the manuscript presented in an intelligible fashion and written in standard English?

Reviewer #1: Yes

Reviewer #2: Yes

Reviewer #3: Yes

Reviewer #4: Yes

Reviewer #5: Yes

5. Review Comments to the Author

Reviewer #1: The manuscript explores the current status of air quality monitoring (AQM) in urban areas of the Global South (low to middle income countries), its distributional disparities, and its association with economic conditions and political systems. Based on a geocoded dataset of more than 10,000 urban areas, the authors reveal large gaps in the distribution of AQM in the Global South and explore key factors influencing AQM activities. There are a number of issues that need to be revised before the article is published. My comments are below:

1. The article finds that cities in democracies are more inclined to increase AQM activity at high pollution levels, but this correlation does not necessarily imply causation. For example, democracies may themselves have a stronger economic base and public awareness, and these factors may have contributed to both the development of AQM and the establishment of democratic institutions. The article fails to adequately control for these potential confounding factors, thereby weakening the reliability of causal inferences.

2. The article points out that non-democratic countries perform poorly on AQM, but fails to explore how monitoring capacity in these countries could be improved through external pressures or incentives. For example, can the international community promote AQM development in non-democratic countries through technical assistance, financial support, or policy advocacy?

3. The need for the study could be further emphasized by adding to the background introduction some discussion of the trend towards rapid urbanization in the global South and how this trend is exacerbating the air pollution problem.

4.The introduction describes air pollution and the serious threat to public health and the environment. However, the content is not comprehensive enough and should include aspects of climate change, urban thermal degradation, urban sprawl, and ecological stress. It is recommended to refer to the following literature to enhance the comprehensiveness. -The Dynamic Effects of Ecosystem Services Supply and Demand on Air Quality: A Case Study of the Yellow River Basin, China; Investigating the attribution of urban thermal environment changes under background climate and anthropogenic exploitation scenarios; Designing green walls to Designing green walls to mitigate fine particulate pollution in an idealized urban environment; Can green finance improve China's haze pollution reduction? The role of energy efficiency; Building a climate-adaptive city: A study on the optimization of thermal vulnerability; Spatial and Temporal Heterogeneity of Human- Air-Ground Coupling Relationships. Air-Ground Coupling Relationships at Fine Scale; Combined effects of urban forests on land surface temperature and PM2.5 pollution in the winter and summer? summer

5. In the discussion section, it is suggested to further explore why non-democratic countries tend to carry out AQM even when the pollution level is low, which may be related to the political motivation or international pressure of these countries, and adding this discussion can enhance the depth of the study.

6 The article mentions that “non-democratic countries may choose to monitor in less polluted areas”, but does not explore in depth the potential reasons for this choice. It is suggested that additional hypotheses or literature support be added to explain this phenomenon.

7. When discussing future research directions, it is recommended to be more specific and suggest actionable research proposals, such as how to improve monitoring networks using low-cost sensors, or how to assess the fairness of AQM in different socio-economic contexts.

Reviewer #2: This article explores the current state and influencing factors of air quality monitoring (AQM) in cities of the Global South. The research perspective is novel and offers interesting insights for improving air quality governance. My suggestions are as follows:

1.Data-related aspects, How can OpenAQ, WAQI, and PurpleAir implement strict controls over data quality, such as checking data consistency, completeness, and accuracy? How can more detailed analysis and processing of outliers be conducted? In addition to PM2.5, could other pollutant indicators, such as sulfur dioxide (SO₂) and nitrogen oxides (NOₓ), be included to provide a more comprehensive assessment of air quality?

2.Analytical methods, Beyond binary logistic regression and Poisson regression, are there other models that could be considered? Could the interaction between democratic systems and pollution levels be analyzed more deeply, such as exploring how this interaction varies under different levels of economic development?

3.Spatial considerations, Could spatial autocorrelation be taken into account? Spatial statistical methods, such as spatial autoregressive models, could be used to analyze the spatial autocorrelation of AQM activities and explain spatial distribution patterns.

4.Case studies, Could cities with particularly prominent or lagging AQM activities be selected for in-depth case analysis to better understand the influencing factors and mechanisms of AQM activities?

5.Network management, How can AQM networks be constructed and effectively managed to enhance monitoring efficiency and data quality?

6.Policy recommendations, Could concrete improvement suggestions be proposed? Can implementing policies and mitigation measures effectively improve air quality? For example, references could be made to studies on ventilation pathways (10.1016/j.buildenv.2018.09.010, 10.1016/j.scs.2019.101487) and green infrastructure (10.1016/j.scitotenv.2022.155307)?

Reviewer #3: Rewrite abstract focusing on methodology and core results.

State research objectives explicitly and concisely.

Define research gap clearly in a separate paragraph (Lines 84-91). Right now, it reads like a general essay.

The novelty is hidden. Explicitly claim your contribution

Avoid long-winded methodology description. Use tables to summarize datasets and variables

Exclude too much justification like "we are aware..." (Line 217), this belongs to discussion, not methods.

Over-dependence on p-values from permutation test without effect size is weak. Show effect size more prominently.

Discuss limitations more critically

Compare your findings directly with past literature, mention what aligns or contradicts with others.

In conclusion, Practical implications are shallow.

Read these studies to further enrich your study. (https://doi.org/10.1016/j.hazadv.2023.100395) Provides machine learning applications on air quality prediction in a lower-income country. Useful to highlight methodological advancements in data-scarce regions like the Global South. (https://doi.org/10.3390/environments10080141) Discusses urban air pollution modeling in Sri Lanka using ML. Strengthens the argument on challenges of urban AQM in lower-income cities. (https://doi.org/10.1016/j.rineng.2024.102831) Demonstrates SHAP-based explainable AI in environmental monitoring. Can support the significance of public trust in AQM data transparency. (https://doi.org/10.1007/s11356-025-36292-9) Discusses urban environmental monitoring using AI in developing settings. Adds depth to the environmental policy-tech interface discussion.

Reviewer #4: In this paper, the authors propose an innovative approach to monitoring urban air pollution in the Global South. As a result, they have constructed a novel geocoded dataset on air quality monitoring (AQM) in more than ten thousand urban areas in low- to middle-income countries. The authors used the perm package and a Monte Carlo approximation, with 1,000 iterations, to investigate the factors that influence monitoring behavior. To test the hypotheses, the initial strategy involved bivariate analysis. The authors examine variation in air quality management (AQM) practices across cities in the developing world and explore the potential factors that contribute to this variation. The study reveals a significant disparity in AQM implementation in most urban areas in the developing countries. Economic and political conditions are identified as the primary drivers of this discrepancy. While income levels are likely to have a positive effect on AQM standards, the impact of increased air pollution levels may depend on the level of democratic governance. Thus, the article is well written and read with interest. The strength of this research lies in its emphasis on exploring complex and unobvious relationships between democracy, environmental pollution, and gross domestic product (GDP). However, this study does have a number of limitations, the most significant of which is the weak validity of the proposed methodology due to the limited use of ecological and eco-mathematical modelling in the study. The following are recommendations for enhancing the article:

1. The literature review could be enhanced. In particular, it would be beneficial to include research on agent-based modelling of ecological and economic systems within the scope of the review. Within this context, the following significant works should be mentioned and cited:

[1] Akopov A.S., Beklaryan L.A., Saghatelyan A.K. Agent-based modelling for ecological economics: A case study of the Republic of Armenia. Ecological Modelling, Vol. 346, 2017, pp. 99-118,

https://doi.org/10.1016/j.ecolmodel.2016.11.012.

[2] Achilleas Karakoltzidis, Anna Agalliadou, Marianthi Kermenidou, Fotini Nikiforou, Anthoula Chatzimpaloglou, Eleni Feleki, Spyros Karakitsios, Alberto Gotti, Dimosthenis Α. Sarigiannis,

Agent-based modelling: A stochastic approach to assessing personal exposure to environmental pollutants – Insights from the URBANOME project. Science of The Total Environment,

Vol. 967, 2025, https://doi.org/10.1016/j.scitotenv.2025.178804.

2. In the introduction, it would be beneficial to more clearly articulate the purpose of the research and its primary contribution. At present, there is no clear delineation of the scientific innovation of the proposed methodology. Is it the identification of new correlations, the enhancement of the accuracy of monitoring urban air pollution in the Global South, or some other aspect? More precise definitions are required. As mentioned above, it is essential to enhance the component related to the utilization of environmental, eco-economic, and eco-mathematical modelling tools.

3. The section on data and methods should be enhanced. It is necessary to add a subsection on "Ecological Modeling" or "Data Modeling". The most significant of which is the weak validity of the proposed methodology due to the limited use of ecological and eco-mathematical modelling in the study. A concise mathematical representation of the model, which reflects the studied relationships in the data and may be derived from an econometric analysis, would be greatly appreciated.

Therefore, the article needs to be revised.

Reviewer #5: Introduction

[Paragraph 3] You bring up applications of remote sensing and challenges with that, but the discussion needs to be more fleshed out. As currently written, it reads somewhat dismissive of the utility of remote sensing technology. There is a growing momentum and increasing acceptance for what is being termed as ‘Air Quality Monitoring 2.0’, where multiple sources (Legacy AQM, Remote Sensing, Low-cost sensors) are used in a hybrid way to get high-resolution data. There is a lot more acceptance of such monitoring even in the governments, e.g., India. However, even this approach does not eliminate the need for legacy AQM, which is the focus of this paper. Perhaps a better place for a more fleshed-out version of this paragraph could be in the discussion section.

Addition: I see that in the discussion section you have discussed this issue. Perhaps, a slight modification in language here would be sufficient. And useful to include a reference that reviews this new generation of air quality monitoring e.g. https://link.springer.com/article/10.1007/s41810-024-00281-1

[Paragraph 8] Your choice of focusing on urban areas for this study needs to be justified better in the text. You could use supporting arguments like increasing population living in cities worldwide of LMICs, cities are where more population is exposed to poor air quality. When national governments start AQM, they often start with major city/urban areas and then expand; there are a few exceptions, though, where the beginning is near an industrial area away from the city, e.g., certain mining towns in some African countries. You can justify this choice better with relevant citations.

Data and Methods

[Paragraph 1] In ‘Unit of Analysis’ you write “the coordinates of so-called Urban Centres (UC).”. Use of ‘so-called’ here reads very informal. You can replace it with something like “the coordinated of the Urban Centers (UC), as per the OECD definition”; I am assuming that’s the source for you based on Ref #43

[Paragraph 2] Or perhaps you are not using the OECD definition? Are you using the OECD definition and applying it to non-OECD countries? Won’t this be problematic? Have you made any adjustments? This part could use more clarity in your choice of definition of ‘Urban Centres’

Also, this justification of focusing on non-OECD countries is better placed in Introduction section? A bit confusing – is it non-OECD or LMICs (Low-to-upper-middle-income). Are those turn out to be the exact same set of countries?

[Paragraph 4] In ‘Monitoring Stations’ you write “With UCs as the unit of analysis, we merge data on reported terrestrial AQM activity from three different sources with global coverage, monitoring any activity up to mid-April 2024.” Does it mean any AQM data is available up until mid-April 2024? Is there a threshold for duration? Do you expect AQM data to be current?

[Paragraph 5] There are a few countries in Africa where primarily low-cost sensor networks make up the AQM that are financially supported by international philanthropies working towards increasing air quality monitoring. While in some places/cities such projects have formal or informal blessing of governments, in many cases they don’t. Would you not consider that exogenous as well? If not, why? One of your citations, Ref #27, focuses on AQM that are supported by the governments world-wide – a distinction like that could be helpful in your case?

Addition: I see that in the discussion section you have discussed this issue. It will still be worth noting in the methods that when you test a model where you exclude all low-cost sensors, the assumption is that low-cost sensors are not government-operated.

[Paragraph 7] In ‘Economic Resources’, you write “Given that none of these UC has an observable monitoring station, we opt to exclude these 314 entities.” Firstly, typo – UCs*. Secondly, did you consider any alternative approach instead of excluding these 314 entities? Perhaps, looking at nearby pixels and averaging incomes for surrounding pixels?

[Paragraph 9] In ‘Air Pollution’, you describe your choice of dataset, which is remote sensing-based but calibrated to the ground measurements. I believe this is probably the best available choice you had. But won’t the accuracy of these measurements be biased against regions where there is no AQM? Important to mention this bias.

[Paragraph 10] In ‘Conflict’, you write “in over 1000 battle-related fatalities since its inception.” Is this threshold based on similar work elsewhere?

[Paragraph 12] In ‘Other control variables’: Why not use fixed effects for all countries? Why only India and China? You mention the third country Ethiopia also has a significant number of UC-AQM matches?

[Paragraph 13] In ‘Data Analysis’, it will be useful to provide equations for both multivariate analysis models.

[Paragraph 14] In ‘Data Analysis’, you write “To assess the robustness of the results, we use different combinations of decisions throughout the analysis and test them using multiverse analysis.”. Replace ‘decisions’ with ‘assumptions’?

Results

[Figure 1] It would be helpful if you also assigned different marker types with colours for different databases. It is a good practice to make the figures friendly to colour-blind readers and also advantageous if printed in black-and-white.

[Paragraph 3] You write “Less polluted cities are more likely to be monitored, and cities in nondemocratic settings tend to be more polluted (p < 0.01 & p < 0.01, permutation tests).” This could also be interpreted as the cities have lower pollution because they were monitored.

[Figure 4] So the effect sizes are in numerical values? The dependent variable is the Presence of AQM, correct? What is the threshold for the classification in this logistic regression? It’d be useful to connect caption to your multivariate regression equation provided earlier.

[Paragraph 8] You write “Although the effect is weaker for reference-grade monitors, it remains statistically significant when applying what we consider the most appropriate estimation strategy (Fig. 6).” I am assuming you are referring to the combination of ‘Everything – embassies’ with ‘IND/CHN subsampled’. Calling it ‘best estimation strategy’ is misleading. You can refer to it as your baseline model – or something to that order – and refer it like that throughout the paper. Earlier, when you mentioned decisions, you can refer to it as the baseline model again.

Discussion

Your special treatment of India and China, justifiably so due to the high number of urban centers and more AQM data, makes me curious about AQM journey of these two countries. A simple search on Google Scholar results in several studies that examine AQM status/drivers/history for these two countries. Perhaps synthesizing some of them and including a brief discussion of how those findings may align with your findings will enrich this discussion. Quantitative analysis that relies on proxies of complex socioeconomic parameters (e.g., democracy) to find causal mechanisms of policies (here, AQM) gets more grounding when compared to findings from qualitative research.

The following citations may be useful in this case:

https://doi.org/10.1016/j.jenvman.2019.110031 - history of monitoring in China

https://doi.org/10.1016/j.jenvman.2021.113232 - an analysis similar to yours focussed on cities in China

https://doi.org/10.1016/j.envc.2021.100431 - history of monitoring in India

6. PLOS authors have the option to publish the peer review history of their article (what does this mean?). If published, this will include your full peer review and any attached files.

Reviewer #1: No

Reviewer #2: No

Reviewer #3: No

Reviewer #4: No

Reviewer #5: **Yes: **Viraj Sawant

---

## [Author Response · Author response to Decision Letter 1]

29 Oct 2025

PONE-D-25-11715 Response Memo

“Large Gaps in Monitoring Urban Air Pollution in Low- and Middle- Income Countries Associated with Economic Conditions and Political Institutions”

Dear editorial team of PLOS ONE, dear reviewers,

Many thanks for all your very helpful comments and suggestions. We have thoroughly revised the manuscript in response to these comments and suggestions, and we are pleased to herewith resubmit the revised manuscript “Large Gaps in Monitoring Urban Air Pollution in Low- and Middle- Income Countries Associated with Economic Conditions and Political Institutions” (PONE-D-25-11715).

In this memo, we respond point-by-point to all reviewer and editor comments, and also submit one version of the revised manuscript with track changes, along the manuscript without track changes.

We believe that the paper has benefited very much from your comments and suggestions, and the resulting revision. We greatly appreciate the opportunity to revise the manuscript for consideration at PLOS ONE and look forward to your response. It goes without saying that we are very much willing to undertake any further revisions you might deem necessary in order to get the paper published in PLOS ONE .

In addition to the point-by-point response, please note that we have slightly edited all figures from the manuscript because of a small update to the AQM dataset. Some locations were wrongfully filtered out and others were duplicated. We carefully checked whether this led to significant changes in our interpretation of the findings, and adjusted when necessary, which only concerned the mentioning of descriptives in the results section.

We also made sure the countries of interest for our study were referred to, and defined consistently throughout the document, and edited title, abstract, and manuscript to refer to low- and middle- income countries (LMIC). Also, because Figure 6 and (former) Supplementary Figure 4 covered all combinations of the data displayed in (former) Supplementary Figure 3, we deleted (former) Supplementary Figure 3.

The authors

Reviewer #1:

The manuscript explores the current status of air quality monitoring (AQM) in urban areas of the Global South (low to middle income countries), its distributional disparities, and its association with economic conditions and political systems. Based on a geocoded dataset of more than 10,000 urban areas, the authors reveal large gaps in the distribution of AQM in the Global South and explore key factors influencing AQM activities. There are a number of issues that need to be revised before the article is published. My comments are below:

1. The article finds that cities in democracies are more inclined to increase AQM activity at high pollution levels, but this correlation does not necessarily imply causation. For example, democracies may themselves have a stronger economic base and public awareness, and these factors may have contributed to both the development of AQM and the establishment of democratic institutions. The article fails to adequately control for these potential confounding factors, thereby weakening the reliability of causal inferences.

Response #1.1:

Thank you very much for flagging this limitation of the paper. We fully agree that identifying causal effects of democracy is impossible because democracy cannot be manipulated experimentally or quasi-experimentally in any meaningful study design. We have revisited the paper and adjusted its wording to make it very clear that we talk about association or correlation, and not causation. Regarding the issue of confounders/control variables, we have probably not made it clear enough that our models do control for various confounding factors. For instance, we include income levels alongside democracy scores in every model, and also look at the combined (interactive) effect of income and democracy. To our knowledge, there is no reliable data for public awareness at the city or country scale that we could use to control for this potential confounder, given that we compare around 10000 cities. However, as we include pollution levels, income levels, and democracy levels in all models we believe that these factors do at least to some extent, indirectly, proxy for awareness levels, because many micro-level studies show that individuals with higher incomes living in more polluted settings with higher levels of democracy (and thus also higher levels of freedom of science and the press) tend to be more aware of environmental problems. One (albeit rather indirect) proxy for awareness we considered is The Worldwide Governance Indicators’ Voice and Accountability. But this measure is highly correlated with our democracy score (Pearson’s R = 0.956), meaning that it makes no sense to include it in our models. In other words, we believe that our models now include most of the confounders that, from a theoretical perspective, might also affect monitoring behaviour, and for which we can obtain data of reasonable quality and scope. However, we are very open to including further confounders the reviewer might have in mind, and for which there is data.

2. The article points out that non-democratic countries perform poorly on AQM, but fails to explore how monitoring capacity in these countries could be improved through external pressures or incentives. For example, can the international community promote AQM development in non-democratic countries through technical assistance, financial support, or policy advocacy?

Response #1.2:

We fully agree that this is a very interesting and important question. One might hypothesize that in lower income contexts AQM capacity can be increased mostly via country-external support, and that such support is more likely to materialize and make a difference in more democratic settings. Our empirical data analysis part as such cannot provide an answer to this question as this would require city-level data on how much foreign assistance for AQM is flowing into which city and what the effects of it are. To flag this important issue, we added the following text to the discussion section: “Country-external governmental or non-governmental actors could catalyze more AQM via technical assistance (e.g. calibration, open-data systems), co-financing of reference backbones and low-cost densification of monitoring networks. They could also create more pressure on policy-makers to make monitoring results public and allow for independent audits. One interesting hypothesis to study in further research could be whether cities in more democratic contexts are more likely to receive such assistance and make more efficient and effective use of it.”

3. The need for the study could be further emphasized by adding to the background introduction some discussion of the trend towards rapid urbanization in LMIC and how this trend is exacerbating the air pollution problem.

Response #1.3:

Thank you for pointing this out. We have added this text to the introduction: “[...] These problems are amplified by the fact that urban air pollution coincides and interacts with climate change and other ecological stressors, which implies that substantial synergies and co-benefits could be achieved from pursuing both clean air and climate mitigation measures (Cao et al. 2025, Dong et al. 2022; Vandyck et al. 2018).”

4. The introduction describes air pollution and the serious threat to public health and the environment. However, the content is not comprehensive enough and should include aspects of climate change, urban thermal degradation, urban sprawl, and ecological stress. It is recommended to refer to the following literature to enhance the comprehensiveness. -The Dynamic Effects of Ecosystem Services Supply and Demand on Air Quality: A Case Study of the Yellow River Basin, China; Investigating the attribution of urban thermal environment changes under background climate and anthropogenic exploitation scenarios; Designing green walls to Designing green walls to mitigate fine particulate pollution in an idealized urban environment; Can green finance improve China's haze pollution reduction? The role of energy efficiency; Building a climate-adaptive city: A study on the optimization of thermal vulnerability; Spatial and Temporal Heterogeneity of Human- Air-Ground Coupling Relationships. Air-Ground Coupling Relationships at Fine Scale; Combined effects of urban forests on land surface temperature and PM2.5 pollution in the winter and summer? summer

Response #1.4:

Thanks for pointing this out. If the editor allows (in view of editorial guidelines, including limits on references), we would like to add more references to emphasize how air pollution also relates to other urban environmental degradation issues. See also Response 1.3

5. In the discussion section, it is suggested to further explore why non-democratic countries tend to carry out AQM even when the pollution level is low, which may be related to the political motivation or international pressure of these countries, and adding this discussion can enhance the depth of the study.

Response #1.5:

This comment is very similar to the next one, so we have addressed both points under response #1.6.

6 The article mentions that “non-democratic countries may choose to monitor in less polluted areas”, but does not explore in depth the potential reasons for this choice. It is suggested that additional hypotheses or literature support be added to explain this phenomenon.

Response #1.6:

Thanks a lot for noting this. We also found this result very interesting. At this stage, the finding must remain an empirically observed pattern. Systematically identifying the potential causes for this would probably require qualitative interviews with key officials in (hundreds of) such cities, which is far beyond the scope of our paper. However, one candidate explanation could be that policy-makers in less democratic settings have more leeway to place monitoring devices in (relatively) less polluted locations in order to make the city’s pollution situation look better than it actually is. In the discussion section, we flag this hypothesis as worthy of further research. In any event, differences between reference-grade and low-cost sensor prevalence we observe suggest that in non-democracies, governments set up one reference-grade station and commonly leave it at that, whereas low-cost sensors proliferate more in highly polluted areas of the more democratic LMIC.

We revised the discussion section as follows:

“The observation that non-democracies are more likely to monitor in lower pollution areas suggests that non-democracies’ monitoring locations may be biased by political considerations. While non-democracies may establish monitors in cleaner places in response to international pressures, democracies may prioritize highly polluted cities and establish more extensive monitoring networks there due to electoral incentives to provide public environmental goods. It is worth noting, however, that although environmental agencies in democracies generally operate with greater autonomy, instances of strategic misreporting have been documented in both the US and in China (Grainger and Schreiber 2019; Mu et al. 2024).”

In essence, while the research presented here can identify general patterns in the democracy-AQM relationship, the causal mechanisms driving particular patterns require further research. “

7. When discussing future research directions, it is recommended to be more specific and suggest actionable research proposals, such as how to improve monitoring networks using low-cost sensors, or how to assess the fairness of AQM in different socio-economic contexts.

Response #1.7:

Absolutely. We fully agree and added more specific suggestions on how different research designs could help answer important questions about AQM. For instance, we added the following text: “This would require finer grained data than what was used in this paper to capture within-city variations, and could be coupled with qualitative interviews to describe the process of AQM placement. [...] Experimental study designs, such as RCTs, randomizing the placement of low-cost monitors could help us understand the impact that more localized air quality information may have on people’s awareness, concern, behavioral changes to reduce pollution exposure, and policy-preferences and demands.”

We also added a citation (discussion) pointing to a recent review, discussing avenues for low-cost sensor networks (Bagkis et al. 2025). In addition, we also added various suggestions for further research as noted in the responses to the comments further above.

Reviewer #2:

This article explores the current state and influencing factors of air quality monitoring (AQM) in cities of the Global South. The research perspective is novel and offers interesting insights for improving air quality governance. My suggestions are as follows:

1.Data-related aspects, How can OpenAQ, WAQI, and PurpleAir implement strict controls over data quality, such as checking data consistency, completeness, and accuracy? How can more detailed analysis and processing of outliers be conducted? In addition to PM2.5, could other pollutant indicators, such as sulfur dioxide (SO₂) and nitrogen oxides (NOₓ), be included to provide a more comprehensive assessment of air quality?

Response #2.1:

Thank you for raising this important issue. Each data provider on whom we rely aims at removing inaccurate measurements and identifying consistently defect monitors. OpenAQ’s recent API updates add the option of downloading the number of valid measurements compared to the expected number of measurements for such a sensor (https://docs.openaq.org/resources/measurements#summary). WAQI uses data from nearby instruments to calibrate and identify outlier measurements and filter them out (discussion with an engineer from WAQI, and their website states https://aqicn.org/contact/ “a set of real-time AI algorithms are used to detect abnormal data conditions (sparks, low reporting, etc.) and automatically 'disable' data reported from defective stations.” . Additional analysis of air pollution measurements’ accuracy would require significant data processing from all these sources, which is far beyond the scope of our present research. Thus, we rely on the provider’s quality control for the identification of defect monitors. As an additional precautionary measure, we only include monitors that were active for more than 30 days in OpenAQ or PurpleAir, or that were active for at least 3 weeks in the month of April 2024 for WAQI. We added a clarification of this to the Methods section, which mirrors the reviewer’s comment here: “Data providers like WAQI and OpenAQ use data validation algorithms to identify outliers and abnormal measurements. As an additional step, we exclude monitors that were reporting for less than 30 days, and those that reported for less than 2 weeks on the WAQI API in the month of April 2024.” Please note, however, that the purpose of our study is not to explain pollution levels (for which we would have to be far more careful about the quality of reported pollution levels), but simply to explain whether cities do or don’t have any observable AQM. And even with our rather crude definition of this outcome variable, we can detect such activity in only around 10% of all cities in our sample.

Additionally we added the following sentence to the discussion: “[...] Future work could also examine how platforms like OpenAQ, WAQI, and PurpleAir are improving the transparency and robustness of their data quality practices, including outlier detection and handling.

2.Analytical methods, Beyond binary logistic regression and Poisson regression, are there other models that could be considered? Could the interaction between democratic systems and pollution levels be analyzed more deeply, such as exploring how this interaction varies under different levels of economic development?

Response #2.2:

Thank you very much for the suggestion to consider alternative statistical estimation strategies. We used logistic and Poisson regressions as they allow us to model both the presence of a monitor, and the number of monitors in each city, or country. These are the state-of-the-art estimation

---

## [Decision Letter · Decision Letter 1]

17 Dec 2025

Large gaps in monitoring urban air pollution in low- and middle- income countries associated with economic conditions and political institutions

PONE-D-25-11715R1

Dear Dr. Bernauer,

We’re pleased to inform you that your manuscript has been judged scientifically suitable for publication and will be formally accepted for publication once it meets all outstanding technical requirements.

Kind regards,

Floris Vermeulen

Academic Editor

PLOS One

Additional Editor Comments (optional):

Reviewers' comments:

Reviewer's Responses to Questions

**Comments to the Author**

1. If the authors have adequately addressed your comments raised in a previous round of review and you feel that this manuscript is now acceptable for publication, you may indicate that here to bypass the “Comments to the Author” section, enter your conflict of interest statement in the “Confidential to Editor” section, and submit your "Accept" recommendation.

Reviewer #5: All comments have been addressed

2. Is the manuscript technically sound, and do the data support the conclusions?

Reviewer #5: Yes

3. Has the statistical analysis been performed appropriately and rigorously?

Reviewer #5: Yes

4. Have the authors made all data underlying the findings in their manuscript fully available?

Reviewer #5: Yes

5. Is the manuscript presented in an intelligible fashion and written in standard English?

Reviewer #5: Yes

6. Review Comments to the Author

Reviewer #5: Authors have addressed the concerns raised in the first review adequately. They have made appropriate additions to text where it was pointed out that the discussion is insufficient.

7. PLOS authors have the option to publish the peer review history of their article (what does this mean?). If published, this will include your full peer review and any attached files.

Reviewer #5: No

---

## [Editor Report · Acceptance letter]

PONE-D-25-11715R1

PLOS One

Dear Dr. Bernauer,

I'm pleased to inform you that your manuscript has been deemed suitable for publication in PLOS One. Congratulations! Your manuscript is now being handed over to our production team.

Kind regards,

on behalf of

Dr. Floris Vermeulen

Academic Editor

PLOS One